# Reading through the eyes of a university student: A double-masked randomised placebo-controlled cross-over protocol investigating coloured spectacle lens efficacy in adults with visual stress

Darragh Liam Harkin [ID]°*, Julie-Anne Little°, Sara J. McCullough°

Centre for Optometry and Vision Science, School of Biomedical Sciences, Faculty of Life and Health Sciences, Ulster University, Coleraine, Northern Ireland

° These authors contributed equally to this work.
* harkin-d13@ulster.ac.uk

## Abstract

### Background

Visual stress is a reading disorder characterised by perceptual distortions, asthenopia and headache whilst reading, alongside increased sensitivity to repeated striped patterns ('pattern glare'), in the absence of underlying ocular pathology. Coloured filters including tinted spectacle lenses and coloured overlays/acetates have been reported to ameliorate visual stress symptoms. However, evidence on coloured spectacle lenses efficacy at managing symptoms of visual stress, particularly in adults, is lacking, with recent systematic reviews advocating the need for large-scale randomised control trials.

### Methods

This is a double-masked randomised placebo-controlled superiority trial. University students identified with symptoms of visual stress, through use of a reading symptom questionnaire and mid-spatial frequency pattern glare test, will be recruited. Sample size for power of 90% at 5% significance, accounting for 10% dropout will be 65. Participants will be randomly assigned experimental and control coloured spectacle lenses to wear for six weeks followed by a two week washout period, prior to wear of the alternate lenses for a further six weeks with a two week washout period. Participants will compare both sets of spectacle lenses in a 'head-to-head' comparison after the secondary washout period, prior to choosing the preferred lenses for voluntary future wear. Long-term adherence to the preferred lenses will be assessed three months post-comparison. Researchers and participants will be masked to spectacle lenses worn throughout the duration of the trial. Reading performance will be

[🔓] OPEN ACCESS

**Data availability statement:** No datasets were generated or analysed during the current study. All relevant data from this study will be made available upon study completion.

**Funding:** Materials (spectacle lenses & spectacle frames) have been funded by the Local Ophthalmic Committee Central Optical Fund (Ref: 202302). Additionally, DH is funded by Department for the Economy, Northern Ireland with a PhD scholarship at Ulster University. Funders have not and will not have a role in the design nor implementation of the trial protocol. Funders have not been involved in the writing of or decision to publish this protocol, nor shall they be involved in the analysis for dissemination of this trial's findings.

**Competing interests:** The authors have declared that no competing interests exist.

assessed with both sets of lenses at various time points within the trial. A range of reading tests, reading symptoms and pattern glare evaluation will be used to monitor change in reading performance and visual stress symptoms during the trial.

## Discussion

The study will evaluate the hypothesis that coloured spectacle lenses increase reading speed and reduce severity and frequency of reading symptoms in adults with visual stress.

## Trial registration

The trial is registered at ClinicalTrials.gov: NCT04318106.

---

## Introduction

Visual Stress, occasionally referred to as 'Meares-Irlen Syndrome' and 'Scotopic Sensitivity Syndrome' is a visuoperceptual reading disorder with a reported prevalence varying between 13–96% [1] in both adults and children. Variable prevalence can be explained by the array of symptoms and signs employed to diagnose the condition, sample population tested [2,3] and unclear evidence on the underlying aetiology. Attempts have been made to standardise diagnostic criteria [3–6] however these have not been validated. Symptoms of visual stress can be broadly described as the presence of perceptual distortions, such as words and letters 'moving' or 'merging', and appearance of 'rivers' or coloured patterns within lines of crowded text; typically black text printed on white backgrounds [3,6]. There are several aetiologies proposed to cause visual stress but, in brief, these either fall under (i) A deficit in magnocellular processing and inability to process high temporal frequency 'flicker' [7,8] or (ii) Hyperexcitation of the visual cortex from repetitive black-on-white striped patterns often found in written text [9,10]. Colorimetry – the practice of determining preferred chromaticity of coloured spectacle lenses to assist with reading symptoms, is performed in some primary eyecare settings. Coloured filters, in the form of tinted spectacle lenses or sheets of coloured acetate have been reported to improve reading performance and reduce symptoms in individuals with visual stress [5,11–13] in previous randomised controlled trials and case-controlled studies.

With regard to theory (i), coloured filters are thought to supress potentially aberrant intraretinal image patterns from saccadic eye movements [7,14] and improve accuracy of ocular movements whilst reading [15] in symptomatic individuals with underlying magnocellular pathway deficits. For theory (ii) coloured filters are proposed to reduce the extent of cortical hyperexcitation by attenuating the amount of potentially aversive visual information that is received by a hyperexcitable primary visual cortex [16,17].

A few studies have investigated whether coloured lenses reduce hyperexcitability in the visual cortex associated with visual stress and/or migraines using functional neuroimaging and visual evoked potentials [16,18,19]. While some studies support a

reduction in the cortical excitability with coloured filters [16] others report conflicting results [19] or only reductions in those with co-existing migraines or headaches [18]. Further work is required to fully understand the aetiology and treatment mechanisms of visual stress and what areas of the visual cortex are specifically hyperexcitable in those with visual stress.

Yet, to date investigations into the efficacy of coloured lenses at ameliorating symptoms of visual stress are scant. Recent systematic reviews conclude that a large-scale double-masked randomised control trial needs to take place to investigate coloured lens efficacy in subjects with visual stress [1,20–22]. Two meta-analyses further highlight the necessity for high quality evidence to strengthen the knowledge base in relation to coloured lens efficacy for those with reading difficulties [23,24]. Furthermore, there is some discrepancy in terms of the clinical versus statistical significance of findings of coloured spectacle lenses efficacy among previous studies. This is demonstrated by the use of percentage reading speed change noted with optimal coloured filter to determine efficacy of coloured lenses, with percentage reading speed change with optimal filters reported as ranging between 0.90% (p = 1.0) [13] and 6.4% (p = 0.03) [12].

Additionally, professional optometric bodies' management guidance in relation to the practice of colorimetry is guarded, citing a lack of robust evidence to either endorse or reject the practice [25–27]. This protocol will detail the methods employed in a double-masked cross over randomised placebo-controlled trial. This superiority trial aims to test the hypothesis that providing coloured lenses to those with symptoms of visual stress will be effective at improving reading performance and reduce frequency of symptoms experienced whilst reading.

## Methods and materials

### Study design

This study is designed as a double-masked cross-over randomised placebo-controlled trial of coloured lens efficacy in adults with symptoms of visual stress. The trial design is supported by the Standard Protocol Items: Recommendations for Intervention Trials (SPIRIT) framework (see supporting information S1 File). Participants will be evenly and randomly assigned one of two sets of coloured spectacle lenses for initial spectacle wear for a period of six weeks, followed by wear of an alternate pair of spectacle lenses for a further six weeks. Coloured spectacle lenses will differ in their chromaticity, with the 'experimental' coloured lenses, tinted to optimal chromaticity, which are presumed to reduce perceptual distortion symptoms, whilst 'control' lenses will comprise a chromaticity similar in perceptual colour appearance to 'experimental' lenses, to assist with masking, but differing in chromaticity, such that visual stress symptoms are neither improved nor worsened with use of tint. The 'Control' lenses shall therefore act as a placebo. Researchers and participants will both be masked to the spectacle lenses worn. A two week washout between periods of spectacle lens wear will be employed, so that any potential legacy effects of coloured lenses are negated.

### Study population

**Sample size calculation.** The Wilkins Rate of Reading Test reading speed has been used as the outcome measure for an improvement with use of coloured filters [12,13,28–31]. Sample size was calculated using statistical software (G-Power, Version 3.19.4, 2024), for an a priori within-factors repeated measures ANOVA of the Wilkins Rate of Reading Test (the primary outcome measure of this trial). A partial eta squared ($\eta\rho^2$) of 0.0355 was used with a conservative small to medium effect size ($f$) of 0.1985. Probability of a type-1 error ($\alpha$), of 0.05 and power ($1-\beta$) of 0.90 were applied. Correlation among repeat measures was assumed 0.5 and nonsphericity correction ($\epsilon$) was held at 1.0. A sample size of 59 participants was determined appropriate for this investigation The required sample size has been increased by 10% to account for attrition. Therefore, 65 participants shall be recruited for the trial.

**Study location and participant demographics.** Participants will be recruited from the undergraduate student population at Ulster University, Coleraine, Northern Ireland, UK and all study visits will take place at the Centre for Optometry and Vision Science at Ulster University. An undergraduate student population has been chosen as an appropriate homogenous population to trial coloured spectacle lenses, as it is expected that this cohort will spend greater

periods of time reading than the general adult population. Therefore, any potential benefit of a precision tint can be trialled in a population who will engage heavily with reading material.

**Inclusion criteria.** All participants shall be adult undergraduate students at Ulster University, aged 18–45-years. Participants shall be recruited in their first year of their current university degree. It is anticipated that the majority of participants will be young adults between the ages of 18–25, owing to the high prevalence of this age cohort attending undergraduate lectures. All included participants will be pre-presbyopic, i.e., not requiring a specific refractive error addition for near vision tasks/reading. Participants will be enrolled in the trial on the basis of presence of symptoms suggestive of visual stress, as determined by a reading symptom questionnaire and as detailed in Table 1.

**Exclusion criteria.** Participants will be excluded if they satisfy any one of the following criteria:

- Habitual distance monocular acuity poorer than 0.1 logMAR.

- Habitual near monocular acuity poorer than N5 at 40 cm.

- Presence of manifest strabismus and/or decompensating phoria observed on alternating cover test.

- Near point of convergence greater than 10 cm.

- Failure to meet Duane-Hoffstetter Criteria for minimum binocular push-up accommodative amplitude (>15-[0.25]*[age in years]) – see Monger et al. [32]

- Absence of observable stereopsis on the Randot stereotest.

- Formal diagnosis of photosensitive epilepsy

Diagnosis of photosensitive epilepsy will form an exclusion criterion owing to the epileptogenic nature of the repeated striped patterns of the mid-spatial frequency 'pattern 2', of the pattern glare test, as detailed in the tests instructions [33].

**Table 1. Recruitment criteria used to determine presence of visual stress.**

| Participants must satisfy at least one of the following sets of inclusion criteria: | |
|---|---|
| 1. Presence of Significant Symptoms and Pattern Glare | Presence of any two symptoms with frequency scores ≥3, from the following:<br>'Words/ letters look jumbled'.<br>'Patterns seen in the white spaces between words/ letters'.<br>Colours in the white spaces between words/ letters'.<br>'Repeating words/ lines of text'.<br>'Skipping words/ lines of text'.<br>'Losing your place when reading'.<br>'Words/ letters flicker when reading'.<br>'Shimmering over the words/letters'.<br>'Words/ letters wobble when reading'.<br>'Words/ letters move'.<br>'Words/ letters merge'.<br>'Patterns/ shadows seen in text, e.g., rivers'.<br>'Letters/ words stand out in 3D above the page'.<br>'Letters/ words fade or darken on the page'.<br>AND a pattern glare test score >3 |
| 2. Total Symptom Score >50%: | Summed total Likert symptom frequency score of 14-symptoms from 0-5 exceeds a 50% total score threshold (>35/70), when excluding 3 of the 17 Likert symptoms listed in the questionnaire, which may differentially diagnose other ocular abnormality (listed in brackets), which are:<br>'Headaches when reading'. (Headache)<br>'Eyestrain when reading'. (Asthenopia)<br>'Words/ letters look blurred'. (Uncorrected Refractive Error) |

Those diagnosed with dyslexia and/or other specific learning disability (SpLD) will not be excluded from the trial owing to the concurrence of these conditions and visual stress [5], and differences in the symptom profiles and diagnostic criteria used to determine presence of reading difficulty. Irrespective of dyslexia/SpLD diagnosis, researchers will assume presence of VS in all identified participants on the basis of diagnosable pattern glare, concurrent with reported visuoperceptual reading symptoms, as detailed in Table 1. Data on profile of SpLD/dyslexia diagnosis of participants will be available to researchers after trial completion and may be considered in post-hoc analysis of coloured lens efficacy.

Further binocular vision tests which will be conducted at baseline, but are not used as exclusion criteria for the clinical trial include: fusional reserves (positive, negative and vertical); near associated phoria; accommodative facility; dynamic retinoscopy and calculation of the AC/A ratio.

## Recruitment

**Reading symptoms screening.** A reading symptom questionnaire was developed to screen large cohorts of undergraduate students in lecture theatres (see Supporting Information S2 File). There are currently no validated criteria used to diagnose visual stress [1,3]. Therefore, reading symptoms employed in the questionnaire were devised from symptoms of visual stress reported in the literature such as those proposed by a Delphi study [3] and the 'Visual Stress Index' [6]. The questionnaire includes 17-items and 6-point Likert scale with descriptors of symptom frequency (0=never; 5=always). Participants will be directed to consider their reading symptoms on a day-to-day basis when completing the questionnaire. Additionally, participants will be provided with a sample of crowded text in a foreign language, devoid of context, to consider when answering the questionnaire, if they are uncertain about the explanation of a particular reading symptom [6].

Questionnaire test-retest reliability was assessed on a sub-group of 115-undergraduate students through repeat issuing of the questionnaire six weeks after initial screening. Participants were blinded to motivation for retesting. High intraclass correlation coefficients were noted (0.91), with low bias of −1.99 (change in symptom scores with repeat measures, with a negative value indicating an increase on total summed symptom score on repeat) by Bland-Altman analysis [34]. Overall questionnaire internal consistency was also considered high by Cronbach's alpha ($\alpha=0.92$) calculations. These findings confirm overall questionnaire test-retest acceptability for screening purposes.

Further questions exploring participants' ocular and general health will be posed, as well as questions investigating presence of potential visual stress co-morbidities such as; headache [35,36]; formal diagnosis of migraine [37–40]; diagnosis of dyslexia [5,41,42] and/or diagnosis of specific learning difficulties and other general health complaints, known to also be managed with coloured filters [17,43–45].

To determine presence of cortical hyperexcitability, a mid-spatial frequency pattern glare test will also be administered. A 40cm string will be affixed to the test, to measure the proposed viewing distance of 'pattern 2' of the pattern glare test series (spatial frequency 2.3cpd) [33]. Respondents will be directed to circle 'yes/no' responses to each of the seven pattern glare symptoms as per the test instructions [33,46]. Each positive 'yes' response will be summed to determine a total pattern glare score ranging from 0-7.

Reading symptom questionnaires will be administered during term time. The researchers will attend lecture theatres and inform students that they are conducting an investigation of symptoms students experience whilst reading. Recruitment to this study began on 08/12/2022. Repeat rounds of screening will be conducted year-on-year with newly enrolling first-year undergraduate students, if initial recruitment is sub-optimal. It is anticipated that final recruitment will be completed by 20/12/24. Participants will be informed that completion of the questionnaire is voluntary and that all information disclosed will be treated confidentially. Participants will not be informed of the potential to participate in the randomised control trial until after completion of the questionnaire to avoid biasing results.

**Determination of the presence of visual stress.** In the absence of standardised diagnostic criteria for visual stress, recruitment criteria were devised on the basis of previously reported criteria and methods of recruitment to previous

investigations within the literature [5,11,12]. One drawback of previous investigations of coloured filter efficacy is the inability of some studies to accurately determine presence of visual stress in participants, with some incorrectly enrolling participants on the basis of dyslexia diagnosis [47–49] or due to variations in definition of visual stress symptoms (see review by Griffiths et. al. 2016) [1]. The current study will employ strict recruitment criteria to ensure that the most symptomatic individuals are included in the study. Participants who satisfy at least one set of the inclusion criteria as listed in Table 1 shall be invited to take part in the trial. By satisfying either criterion, participants will be recruited on the basis of (i) presence of pattern glare and at least two significant visuoperceptual reading symptoms indicative of visual stress or (ii) overall summed symptom severity across an array of 14-visuoperceptual symptoms.

**Participant invitation.** Once identified with the presence of symptoms suggestive of VS, participants will be contacted by either email or telephone. Participants will be invited to attend a baseline vision assessment at the Ulster University Eye Clinic, conducted by DH. All study visits will be conducted by DH.

## Procedure

**Interventions employed (determination of control lens).** The 'Curve' Intuitive Colorimeter ™ [50] will be used to determine chromaticity of 'experimental' (optimal) and 'control' (sub-optimal) tints, through manipulation of the hue, saturation and luminance of a projected series of three primary colours to comprise a gamut of visible chromaticities spaced throughout the 1976 CIELUV colour space [51,52].

Participants will identify an 'experimental' tint through a series of repeated presentations of opposing hues. Initially, 12-predetermined hues, separated by a 'hue angle' of 30˚ will be presented. Participants will compare the appearance and comfort of a random series of letters forming a striped body of text devoid of context within the Intuitive Colorimeter™, illuminated under white light and presented hue. Participants will note if the presented hue makes the appearance and comfort of the crowded text 'better', 'worse' or 'no-different' than white-light. Beneficial hues will be shortlisted. Saturation of each shortlisted hue will be reduced until symptoms arise. Saturation will then be adjusted to the lowest level where symptoms resolve. A 2-alternative forced choice of preferred hue of ideal saturation will then be presented until an optimal chromaticity can be determined, as previously described [53]. Once optimal hue and saturation are determined, luminance will be attenuated between 'half' and 'full' settings and the participant offered a preference choice. An arc of beneficial therapeutic colours will be uncovered through this technique. This area of therapeutic chromaticities will represent an area on the 1976 CIELUV colour space wherein observers are expected to derive benefit from colour. The authors propose that variation in size and location of this therapeutic region will exist amongst observers. Boundaries of the colour space will be denoted as two 'hue angles' on the Intuitive Colorimeter™ colour wheel, outside which therapeutic benefit is lost. Optimal chromaticity will lie within this colour space. This arc of beneficial chromaticities will be herein referred to as the 'Limit of Therapeutic Effect'.

The 'control' lens will act as the placebo and will be sub-optimal, when compared to 'experimental' tint. To determine control tint, researchers will prescribe a colour that will not be aversive, yet not beneficial to the patient's symptoms of visual stress that lies just outside the 'Limit of Therapeutic Effect'. At the upper and lower boundary of the Limit of Therapeutic Effect, participants will consider if reading symptoms/ visual discomfort experienced are 'worse' or 'no different' compared no colour presented. The hue will be adjusted to when participants confirm that there is 'no difference' when compared to no colour (i.e., not beneficial nor aversive). At these two hues, the researchers propose that these are the candidate control tints. The control tint chosen will be that closest to the optimal tint on the colour diagram, to assist with participant masking – i.e., the apparent colour difference will be more visually alike between control and optimal tint. 'Control' tint will be of the same saturation and luminance to the 'experimental' tint. Researchers are mindful of previous work, stating inter-practitioner and inter-instrumental measurement error may account for ~0.03 units of chromaticity difference (the Euclidean distance between colour coordinates mapped on the 1976 CIELUV Chromaticity Diagram) between lenses [54,55]. Whilst a consistent chromatic separation between lenses will not be applied, to account for assumed difference in

size of 'Limit of Therapeutic Effect' by observers, size of chromatic separation between lenses will be considered in post-hoc analysis of reading performance. The authors appreciate that there is a certain degree of variability in determining sub-optimal tint by this method, as demonstrated in work by Suttle and colleagues (2017) [56], whereby number of just noticeable differences between precision tints varied between 23.8 and 3.1, among 21 observers. However, researchers will aim to ensure that the 'control' tint be inert, through confirmation of participants' visuoperceptual symptoms when viewing the striped body of text within the Intuitive Colorimeter™ illuminated by the assumed 'control' chromaticity. Furthermore, the chromaticity difference, between colour coordinates of 'experimental' and 'control' lenses will be measured by the Mark 2 SpectroCAL Spectroradiometer (Cambridge Research Systems Ltd., Cambridge, UK). Chromaticity coordinates will be plotted by computer software (Light Values Measuring System, Version 6.2.0, Windows 11.0 OS, Jeti Technische Instrumente GmbH Tatzendpromenade 2, Jena, Germany). Chromaticity difference between 'experimental' and 'control' tints will be considered in post-hoc analysis of reading performance with the use of coloured lenses.

Additionally, participants will be graded on their ability to decipher beneficial therapeutic colour and will be classified as 'good', 'moderate' or 'poor' observers in line with previous investigations [55]. Individual descriptors are detailed in Table 2 below. If a participant is unable to derive benefit from any precision tint, they will be excluded from the clinical trial.

**Randomisation and blinding.** Randomisation of interventions will be conducted by the Ulster University School of Biomedical Sciences' Clinical Trials Manager. As participant recruitment will be conducted on an ongoing basis, interventions will be randomised on a simple 1:1 basis without stratification, to ensure that if subpar enrolment is achieved, an equal number of participants will be engaged with each intervention at each timepoint in the trial. Therefore, approximately half of participants will initially receive the optimal tint whilst the other half will firstly receive the sub-optimal tint. Researchers and participants will be blinded as to which lenses are assigned throughout the duration of the trial. Both spectacle frames housing the spectacle lenses will be identical to mask researchers and participants to the intervention employed.

**Spectacle wear periods and washout.** Participants will wear each intervention for a period of six weeks. Participants will be informed that spectacle wear is voluntary and to wear as much as desired. Participants will be advised to avoid wearing coloured spectacle lenses when driving at night as per manufacturer's specification [52], to ensure that spectral transmission complies with College of Optometrists' Guidance for Professional Practice on provision of coloured filters when driving [57]. In the unlikely event of adverse event related to spectacle wear whilst driving, incident(s) will be monitored and reported to the project's chief investigator SJM. Within the first and final week of spectacle wear, participants will be asked to complete a spectacle wear diary online (Research Electronic Data Capture, version 11.0.3, 2021, or paper copy where necessary) at the end of each day of wear. The diary will record the number of hours the spectacles are worn daily. If the participant indicates that they have not worn the spectacles, they will be asked to provide reasons for this. Participants will also be asked to note the number of hours completing concentrated and/or near-work tasks and if the coloured spectacles have been worn to assist with these tasks. Participants will also be asked to grade how helpful (if at all), the spectacle lenses have been in assisting their reading performance on a six point Likert scale (0 = 'Not at all helpful' to 5 = 'Extremely helpful'). Fig 1 comprises a flowchart of the questions that will be posed to participants in their online spectacle wear diary.

**Table 2. Grading scales of participants' responses on the intuitive colorimeter™ adapted from Aldrich et al (2018) [55].**

| Grading | Description |
|---|---|
| Good | Participant identifies an optimal colour and does not deviate |
| Moderate | Participant identifies a colour with only minimal adjustment |
| Poor | Participant is unsure and/or colour is repeatedly adjusted |

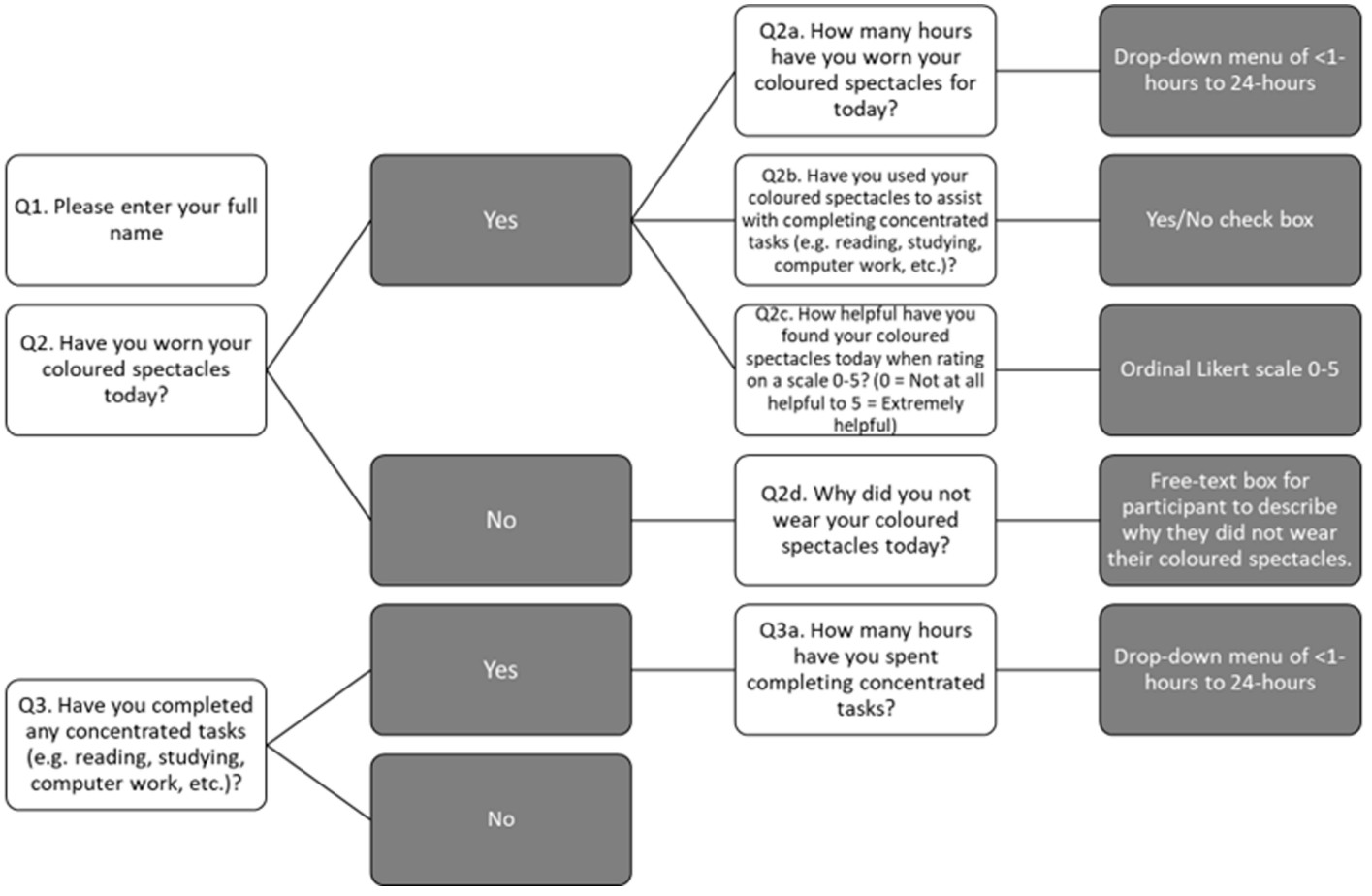

**Fig 1. Flowchart of spectacle wear diary questionnaire.**

After six weeks of wear, a two week washout period will be initiated. Participants' spectacles will be collected and stored securely. Washout will ensure potential carryover effects from initial spectacle wear will be eliminated. A two week washout period has been determined appropriate, in line with previous investigations [11,37]. Additionally, research suggests colour memory for a chosen colour is the same after 1-hour as it is after 1-week [58]. An overview of trial visits can be seen in the study flowchart in Fig 2.

Upon completion of wear of both interventions a 'head-to-head' comparison of both coloured spectacles will take place. Participants will observe a crowded passage of text in a foreign language with both pair of spectacles under masked conditions. Participants will be asked to consider the reading symptoms they experience and to determine which pair of spectacles they find the greatest perceived benefit, irrespective of measures of reading speed and performance noted earlier in the trial. Participants will be advised to wear the preferred spectacles voluntarily, as they require. Participants will be contacted via telephone after three-months to assess long-term spectacle wear compliance and to assess if a sustained benefit from coloured lenses can be determined. Unmasking of participants shall not take place until the three-month follow-up to avoid influencing compliance levels in long-term spectacle wear.

**Trial administration.** All data shall be collected by DH, to reduce measurement error as a result of inter-practitioner variability. The trial shall be steered by the principal investigators, SJM, and JAL – two academics with an established history of previous project administration.

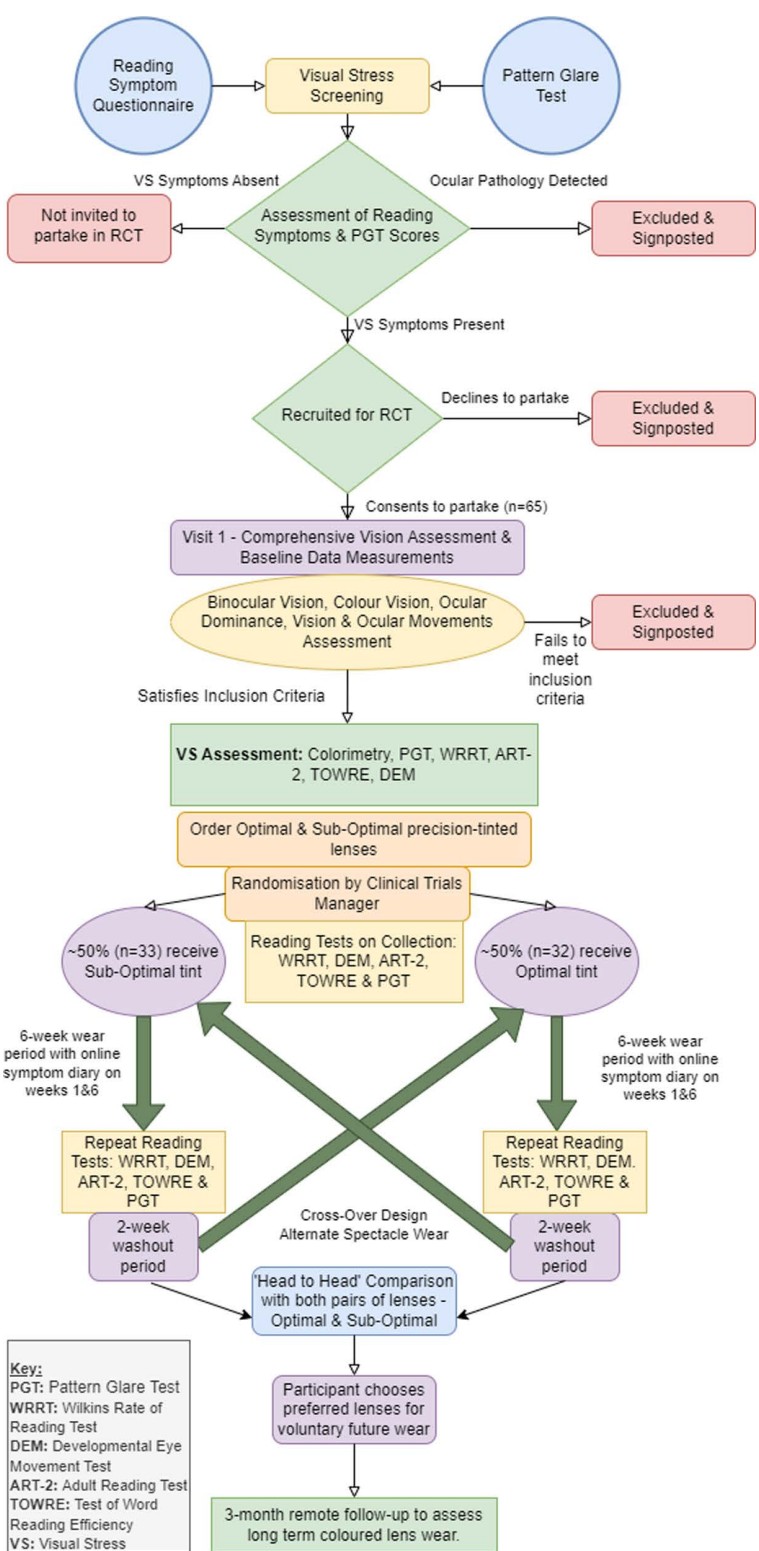

**Fig 2. Randomised control trial efficacy of coloured lenses for visual stress symptoms flowchart.**

## Outcome measures

An overview of the outcome measures employed at each visit during the trial can be seen in Fig 3.

**Primary outcome measure: Wilkins rate of reading test reading speed.** The Wilkins Rate of Reading Test (WRRT) is a long established measure of reading speed with and without the use of coloured filters and has been used in previous reading studies [12,28–31]. As discussed in full by Wilkins et al. [59], the test comprises 15-randomly-ordered monosyllabic commonly used English language words that repeat throughout 10-lines of a size 9-point text to form a crowded paragraph. Owing to the random order of words presented devoid of context, the test isolates the visual input of processing words whilst reading, so that neither syntactic nor semantic reading processes are involved [60].

Participants will read the passage of crowded text aloud within 60-seconds, with and without coloured lenses as detailed in Fig 3 – Schedule of Enrolment, throughout the duration of the trial. The number of words-per-minute (wpm) read correctly will be calculated. During each visit an average reading speed from two measures of the WRRT will be calculated. The examiner will note the number of errors made, namely omissions, reversals, additions, substitutions and mispronunciation [59]. In the event of a ceiling effect whereby all 150 words are read correctly in less than 60 seconds, the time taken to complete the test will be noted, and used to calculate equivalent wpm reading speed [59–61]. The number of additional words read correctly and the percentage reading speed change with coloured lenses will be calculated and compared to baseline measures, in order to assess whether a clinically significant improvement in reading speed has occurred. An increase in reading speed of 15% has been deemed clinically significant by the developer of the Wilkins Rate of Reading test [3]. As percentage change is not normalised, effect sizes and confidence intervals will also be calculated from repeated measures.

**Secondary outcome measures: Symptom improvement, reading speed with further reading tests, pattern glare scores and long-term compliance with coloured lenses. Reading symptom change:** Prior to commencing wear of coloured lenses, participants will be asked to rate reading symptom frequency on the 6-point Likert scale initially used to recruit participants with symptoms suggestive of visual stress to the trial. After 10 minutes and six weeks of coloured lens wear, participants will be asked to repeat the symptom questionnaire. Additionally, after a two week washout period, participants will be asked to rate reading symptom frequency without the use of coloured lenses.

**Developmental eye movement (DEM) test:** The Developmental Eye Movement Test (DEM) Version 1 (Bernell Ltd, 1987), as summarised by Facchin [62] and Tanke et al. [63], was originally proposed to measure horizontal saccadic function of readers [64] but is now thought to more broadly assess oculomotor function whilst reading [62]. Horizontal and vertical reading speed will be measured. Number of omission, addition, transposition, and substitution errors will be noted and will be used to assist calculating adjusted horizontal and vertical reading speeds A ratio of adjusted horizontal:vertical reading speed will then be determined. The DEM will be completed with and without coloured lenses throughout the trial, owing to the low risk of a learning effect from repeat measures of a test comprising a random series of numbers.

**Test of word reading efficiency (TOWRE):** The Test of Word Reading Efficiency (TOWRE) 1st Edition (Pro-Ed Ltd, Austin TX, 1999) comprises a series of two standardised reading tests which independently assess a participant's sight-word efficiency and phonemic decoding ability – as discussed further by Torgeson et al. [65]. Reading speed and number of errors made on both subtests will be recorded. A Total Word Efficiency Standard Score will be determined and compared to age-equivalent normative scores and percentiles [65,66]. Reading tests will be completed with and without coloured lenses as detailed in the Schedule of Enrolment.

**Adult reading test 2nd edition (ART-2):** The Adult Reading Test 2nd Edition (ART-2) (Adult Reading Test Ltd, Hayling Island, 2017) will provide participants with a 'naturalistic' passage of text to be read aloud [67] with and without the use of coloured lenses, as detailed in the Schedule of Enrolment. The test will measure participants' comprehension and reading speed under timed conditions. Both ART-2 Test 1 passages – 'News' (178 words in two paragraphs; Flesch-Kincaid Grade Level 7.9) and 'Wildlife' (179 words in four paragraphs; Flesch-Kincaid Grade Level 7.1), will be alternatively presented to participants to avoid a learning effect. Upon initial collection of 'experimental' and 'control' lenses, participants will be

| | STUDY PERIOD | | | | | | | |
|---|---|---|---|---|---|---|---|---|
| | Enrolment | Allocation | Post-allocation | | | | | Close-out |
| **TIMEPOINT** | $-t_1$ – Visit 1 | 0 | $t_1$ – Visit 2a | $t_2$ – Visit 2b | $t_3$ – Visit 3a | $t_4$ – Visit 3b | $t_5$ – Visit 4 | $t_x$ – Visit 5 |
| **Visit Summary** | *Baseline Measures* | *Randomisation & Allocation* | *Spectacle Collection 1* | *Spectacle Return 1* | *Spectacle Collection 2* | *Spectacle Return 2* | *Head-to-Head Comparison* | *Remote 3-Month Follow-Up* |
| **ENROLMENT:** | | | | | | | | |
| *Informed Consent* | X | | | | | | | |
| *Eligibility Screen* | X | | | | | | | |
| *Habitual Vision / VA* | X | | | | | | | |
| *Binocular Vision Assessment* | X | | | | | | | |
| *City University Colour Vision* | X | | | | | | | |
| *Baseline Reading Tests* | X | | | | | | | |
| *Pattern Glare Test* | X | | | | | | | |
| *Colorimetry* | X | | | | | | | |
| *Allocation* | | X | | | | | | |
| **INTERVENTIONS:** | | | | | | | | |
| *'Experimental' Coloured Lens* | | | | | | | | |
| *'Control' Coloured Lens* | | | | | | | | |
| **ASSESSMENTS:** | | | | | | | | |
| *WRRT with Intervention* | | | X | X | X | X | | |
| *WRRT without Intervention* | X | | X | X | X | X | | |
| *PGT with Intervention* | | | X | X | X | X | | |
| *PGT without Intervention* | X | | X | X | X | X | | |
| *TOWRE with Intervention* | | | X | X | X | X | | |
| *TOWRE without Intervention* | X | | X | | X | | | |
| *ART-2 with Intervention* | | | X | X | X | X | | |
| *ART-2 without Intervention* | X | | X | | X | | | |
| *DEM with Intervention* | | | X | X | X | X | | |
| *DEM without Intervention* | X | | X | X | X | X | | |
| *RSQ with Intervention* | | | X | X | X | X | | |
| *RSQ without Intervention* | | | X | | X | | | |
| *Foreign Language Reading Test* | | | | | | | X | |
| *Compliance Check* | | | | | | | | X |

**Key:** VA: Visual Acuity, WRRT: Wilkins Rate of Reading Test; ART-2: Adult Reading Test 2$^{nd}$ Edition; TOWRE: Test of Word Reading Efficiency; DEM: Developmental Eye Movement Test; PGT: Pattern Glare Test; RSQ: Reading Symptom Questionnaire

**Fig 3. Schedule of enrolment, interventions and assessment.**

asked comprehension questions on the completed ART-2 passage, to assess conceptual understanding of the passage. Each set of questions will only be asked once throughout the trial at each spectacle collection appointment, to avoid learning effects.

**Low, Mid and high spatial frequency pattern glare test:** The Pattern glare test (Institute of Optometry (i.O.O) Sales Ltd, London, UK, 2003) will measure presence and severity of cortical hyperexcitation with and without coloured lenses. Participants will be asked to focus on the horizontal striped patterns of increasing spatial frequencies (0.3, 2.3 and 9.4 cpd) at a fixed 40 cm viewing distance as per protocol discussed by Evans & Stevenson [46]. Seven questions relating to perceptual distortion visual symptoms experienced will be posed to participants. Total number of symptoms reported will be summed, where normative values >3 on the mid-spatial frequency grating, may indicate diagnosable pattern glare [46]. The series of pattern glare tests will be repeated throughout the duration of the trial to test the hypothesis that optimally selected coloured filters may reduce severity of pattern glare and thus cortical hyperexcitability.

## Statistical analyses

The Statistical Package for Social Sciences (Version 29.0; IBM Corporation, USA), software will be used for statistical analysis of reading performance and symptom change with use of 'experimental' and 'control' precision tints. Kolmogorov-Smirnov tests will be used to assess whether the data follow a normal distribution pattern. Average-measures of Wilkins Rate of Reading Test reading speed performance with the use of 'experimental' and 'control' lenses will be assessed by repeated measures AVOVA (or the non-parametric equivalent where necessary) testing for both interventions. This will be assessed at three timepoints – baseline measures; initial spectacle collection and after 6-weeks of spectacle wear. Post-hoc analysis using repeated measures ANCOVA testing will be conducted. Covariates in statistical analysis will include: baseline reading speed, the number of errors made in each reading test, number of days spectacle lenses worn, number of hours spectacle lenses worn and number of hours completing concentrated tasks with or without the use of coloured lenses.

Clinically significant improvement noted on the WRRT, will be informed by percentage reading speed improvement with 'experimental' lenses, compared to baseline. To date, no validated repeatability values nor defined clinically significant improvement on the test exists in the literature. Therefore, researchers will classify participants' reading speed improvement by two categories: (i) >5% to 15% improvement; indicating a mild clinical improvement on the WRRT and (ii) >15% improvement; indicating a moderate clinical improvement on the WRRT. These values have been derived from the consensus in the literature, whereby a stricter criterion for reading speed improvement has been applied more recently [3].

Repeated measures ANOVA testing will also assess reading speed on secondary reading tests employed in the trial with both 'experimental' and 'control' lenses, for the Adult Reading Test (2nd Edition), The Test of Word Reading Efficiency (TOWRE) and the Developmental Eye Movement (DEM) test, at baseline, upon initial spectacle collection and after six-weeks of spectacle wear. Additionally, Friedman test will assess the median change of both interventions on low, mid and high-spatial frequency pattern glare test scores. Furthermore, McNemar's Test of Proportions will investigate changes in reading symptom frequency scores at both the start and end of the six week spectacle wear period, with both spectacle lens interventions when compared to baseline reading symptoms reported.

## Data management

Data will be treated and stored in line with the amended Data Protection Act (2018). Reading symptom questionnaires used will have personal information attached to allow the researchers to identify participants with relevant visual stress symptoms, so that they can be contacted to participate in the trial. Once collected and analysed, each questionnaire shall be anonymised and assigned a unique participant identification number. Participants enrolled in the trial will be identified using their identification number assigned to their reading symptom questionnaire to ensure anonymity during data

entry and analysis. All clinical and sensitive information written on hard-copy shall be stored in a secured filing cabinet, of which only the researchers can access. All electronic data will be stored on University-issued encrypted hardware on password-protected and encrypted data analysis programmes. Data management will comply with UK GDPR regulations. All collected data (electronic and hard copy, including data of participants who do not complete the trial) will be held for 10-years in line with university regulations. Anonymised data will be made available upon request upon cessation of the trial.

## Ethical considerations

The protocol (version 2.0; 21/10/22) has been approved by the Ulster University, School of Biomedical Sciences Ethical Filter Committee as of 28/10/2022 (Ref: FC-BMS-189). The trial is currently listed on ClinicalTrials.gov (Ref: NCT04318106). Participants will be informed that participation in the trial is voluntary and that they can withdraw their participation at any time. Informed consent will be ascertained through a written consent form provided alongside the reading symptom questionnaire (see supporting information S3 File– consent form). Risk of adverse incident from coloured spectacle lens wear is low, but will be monitored. In the unlikely event that an adverse incident occurs, it will be reported to the project's chief investigator SJM and through Ulster University Research Governance protocols.

## Trial progress to date

1071 undergraduates at Ulster University have been screened for symptoms of visual stress, 171 undergraduates have met recruitment criteria and have been invited to participate in the trial. 60 participants are currently enrolled in the trial. Researchers plan to screen newly enrolled undergraduate students at the start of the 2024−25 academic year, to ensure adequate trial enrolment.

## Discussion

This study will be the largest randomised control trial investigating coloured lens efficacy in the management of visual stress in adults to date. Previous studies have been criticised for small sample sizes and being underpowered [1], therefore, it has been difficult to draw meaningful conclusions in relation to coloured lens efficacy. Additionally, this randomised control trial will be the first to assess reading symptoms and performance in a homogenous adult population, unlike previous investigations which have included both adults and children [13]. Evidence and guidance in relation to the practice of colorimetry from professional bodies in the United Kingdom is guarded [25,26], advocating for a large scale randomised control trial to take place prior to endorsing the practice. This is echoed in the findings of recent systematic reviews [1,21]. Therefore, we believe this trial is of high importance to inform best clinical optometric practice.

Researchers appreciate that the studied population will be somewhat homogenous in nature (young pre-presbyopic adult undergraduate students). This trial will explore coloured lens efficacy in a cohort of subjects with a relatively equitable baseline level of reading competence, as all undergraduate students enrolled will have to exhibit demonstrable English language competence to enrol at Ulster University. Whilst this will be beneficial at controlling for amount of reading demand each participant is expected to have, irrespective of whether a benefit of coloured spectacle lenses can be rejected or endorsed, the results of this trial may need be replicated in both paediatric and general adult populations, in future investigations. Additional factors such as gender and degree studied will not be controlled for as it there is no evidence to suggest that these factors are significant in relation to visual stress. As invitation to take part will be based on subjective-reported symptom severity, one may assume a certain degree of selection bias will exist. Researchers aim to reduce selection bias through randomisation and masking of participants. Additionally, whilst risk of ascertainment bias cannot be wholly negated, the exclusion criteria for this trial is relatively uncomplicated; ensuring that only those with binocular vision anomaly/amblyopia will be excluded. Furthermore, upon trial completion, researchers will report the number

of participants identified with visual stress, number who attended baseline screening, failed exclusion criteria, dropped out and were lost-to-follow-up.

Irrespective of the results of this trial, we believe that the results generated from this investigation, will prove insightful in informing future optometric practice, whilst expanding the current evidence base on coloured lens efficacy in the management of visual stress. Recent position statements from the College of Optometrists UK [25] and the Association of Optometrists UK [27] suggest weaknesses in the current evidence surrounding the effectiveness of coloured lenses, and that the treatment of visual stress with coloured lenses remains controversial. Yet, over 300 optometry practices throughout the United Kingdom have an Intuitive Colorimeter™ and assess patients with reading difficulties and symptoms of visual stress [68] and the wearing of coloured lenses is not uncommon. In addition, optometric practices may offer alternative methods of visual stress assessment, such as coloured overlay assessment, and 'ChromaGen' contact lens fitting, amongst others. Whilst this study will not directly influence the management of visual stress with these alternative treatments, further knowledge on coloured filter efficacy will be uncovered from the results of this trial. The results of this study will provide the much needed evidence-base for optometrists, orthoptists and other practitioners assessing visual stress and reading difficulties. Results of this trial may steer the future direction of this area of optometric practice in relation to the practice of colorimetry and prescription of precision tinted lenses.

## Supporting information

**S1 File. SPIRIT checklist.**
(DOCX)

**S2 File. Reading symptom questionnaire.**
(DOCX)

**S3 File. Other - Study Protocol.**
(DOCX)

## Acknowledgments

The authors would like to thank study participants for their commitment to take part in the trial. Additionally, the authors would like to thank technical staff at the Centre for Optometry and Vision Science, Ulster University for their support whilst conducting the trial.

## Author contributions

**Conceptualization:** Julie-Anne Little, Sara J. McCullough.

**Formal analysis:** Darragh Liam Harkin, Julie-Anne Little, Sara J. McCullough.

**Funding acquisition:** Darragh Liam Harkin, Julie-Anne Little, Sara J. McCullough.

**Investigation:** Darragh Liam Harkin, Julie-Anne Little, Sara J. McCullough.

**Methodology:** Darragh Liam Harkin, Julie-Anne Little, Sara J. McCullough.

**Project administration:** Julie-Anne Little, Sara J. McCullough.

**Resources:** Julie-Anne Little, Sara J. McCullough.

**Supervision:** Julie-Anne Little, Sara J. McCullough.

**Validation:** Darragh Liam Harkin.

**Writing – original draft:** Darragh Liam Harkin.

**Writing – review & editing:** Julie-Anne Little, Sara J. McCullough.

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
