## [Decision Letter · Decision Letter 0]

29 Sep 2024

PONE-D-24-33918Reading; through the eyes of a university student: A double-masked randomised placebo-controlled cross-over protocol investigating coloured spectacle lens efficacy in adults with visual stress.PLOS ONE

Dear Dr. Harkin,

Thank you for submitting your manuscript to PLOS ONE. After careful consideration, we feel that it has merit but does not fully meet PLOS ONE’s publication criteria as it currently stands. Therefore, we invite you to submit a revised version of the manuscript that addresses the points raised during the review process.

We look forward to receiving your revised manuscript.

Kind regards,

Clara Martínez Pérez

Academic Editor

PLOS ONE

Journal Requirements:

"Materials (spectacle lenses & spectacle frames) have been funded by the Local Ophthalmic Committee Central Optical Fund (Ref: 202302 - URL: https://www.centralfund.org.uk/). As all three authors (D.H.; S.J.M.;  J.A.L.) are involved in the administration of the study, materials funding was awarded to all three authors. S.J.M. as the Prinicipal Investigator, was the primary recipient of this grant. 

Additionally, D.H. is funded by Department for the Economy, Northern Ireland with a PhD scholarship at Ulster University. 

Funders have not and will not have a role in the design nor implementation of the trial protocol. Funders have not been involved in the writing of or decision to publish this protocol, nor shall they be involved in the analysis for dissemination of this trial’s findings."

Reviewers' comments:

Reviewer's Responses to Questions

**Comments to the Author**

1. Does the manuscript provide a valid rationale for the proposed study, with clearly identified and justified research questions?

Reviewer #1: Yes

Reviewer #2: Yes

2. Is the protocol technically sound and planned in a manner that will lead to a meaningful outcome and allow testing the stated hypotheses?

Reviewer #1: Partly

Reviewer #2: Yes

3. Is the methodology feasible and described in sufficient detail to allow the work to be replicable?

Reviewer #1: Yes

Reviewer #2: Yes

4. Have the authors described where all data underlying the findings will be made available when the study is complete?

Reviewer #1: Yes

Reviewer #2: Yes

5. Is the manuscript presented in an intelligible fashion and written in standard English?

Reviewer #1: Yes

Reviewer #2: Yes

6. Review Comments to the Author

You may also provide optional suggestions and comments to authors that they might find helpful in planning their study.

Reviewer #1: The protocol is a longitudinal study of colored lenses in reading for university students. There are numerous statistical deficiencies due to the terseness of the protocol:

1. The randomization procedure is not specified (permuted blocks, complete randomization, e.g.). There is no discussion of the prevention of ascertainment or selection bias.

2. The sample size is based on a treatment difference seen in prior studies. Studies are powered for a CLINICALLY MEANINGFUL DIFFERENCE, not the observed effect size for prior studies.

3. No details are given on what sample size computations are used; since the outcome is based on a repeated measures ANOVA, there need to be measures of variability and covariance in order to compute the sample size. No indication is given as to the distribution of the primary outcome, which appears to be a count or a score based on words.

4. There are no details on the statistical modelling, what model will be used, if any covariates will be examined, what model assessment diagnostics will be used, what modelling assumptions will be made.

5. "Repeat" measures ANOVA should be "Repeated" measures ANOVA.

6. I don't understand the first part of the title, and the punctuation after "Reading".

Reviewer #2: Reading through the eyes of a university student: A double-masked randomised placebo-controlled cross-over protocol investigating coloured spectacle lens efficacy in adults with visual stress.

Harkin et al.

PONE-D-24-33918

Thank you for a well written protocol that helpfully covers most areas of the SPIRIT guidelines. I have several comments that you should consider to improve the manuscript.

Introduction

The three paragraphs are do not follow a logical flow for me. I suggest: visual stress including prevalence and significance, aetiology and evidence supporting proposed mechanisms, treatment options including evidence for and against, professional guidelines, primary aim of trial.

You state “in small scale RCTs” and yet any RCT should be powered to find a difference i.e. scale does not matter. I suggest your statement is unpacked further. For example, do all trials show reduced symptoms and is this statistically and/or clinically significant?

Study Design

What is your argument for a two-week washout period?

One concern here is that there must be some uncertainty in determining a tint that lies just outside the Limits of Therapeutic Effect but also appears similar enough for masking. It would be good to argue this clearly given some (limited) evidence from Suttle et al (2017) that results are variable.

Study Population

Good to argue why you have chosen a student population because it would seem difficult to extrapolate to a wider population affecting the generalisability of your findings. For example, it is reasonable to anticipate an age range clustered around 18-24 with less participants as you go to your upper age limit of 45 (unless you are using stratified sampling). Also, the amount of reading your sample does will be higher than the general population.

Sample size Calculation

Did you assume a RM ANOVA when carrying out this power calculation? At a basic level the SD of the difference is fundamentally important rather than the within subjects SD, which could differ between groups. Could this be given?

Recruitment

This section has sub-sections on screening tests and doesn’t comment on recruitment of participants!

Outcome Measures

Please ensure the exact metric for the outcome measure is reported. And also indicate if it is an average or median measure for example and if any repeats are used.

Why are you using a percentage reading speed change? Percentages are not normalised and relate to different effect size changes.

DEM

You state “will be noted and inform …”. By “inform” do you mean “calculate”? Best to be explicit.

Statistical Analysis

I was unclear about several points here that could usefully be clarified for the reader:

(1) Are you going to report effect sizes and confidence intervals? (good practice)

(2) Are you going to adjust for baseline measures?

(3) Are you going to test for carry over and period effects?

This can affect the analysis. For example, a 2x2 cross-over can be analysed with a t-test on the differences. ANOVA can be useful if you have different time points and want to look at the effect x time interaction (or other anticipated interactions). ANCOVA can be used when adjusting for baseline by including the baseline measures as covariates.

In general, I think the proposed analysis could be explained more fully and more clearly.

Data Management

Make it clear that it’s the (amended) Data Protection Act (2018). If the UK GDPR applies to your data also state you are complying with that.

General

• Is this version 1 of the protocol. Can you report date and version number?

• I see you have a clinical trials manager but do you have a steering committee?

• Worth clarifying that this is a superiority trial

• Are you excluding those with a diagnosis of dyslexia? If not why not?

• Are you matching groups for age, gender etc.?

• Do you have a Data Management Committee? If not state why.

• What will you do with the data for those that don’t complete the trial?

• Do you plan to monitor adverse events? For example, what if (worst case) a participant wore the intervention while driving and had a RTA?

7. PLOS authors have the option to publish the peer review history of their article (what does this mean? ). If published, this will include your full peer review and any attached files.

**Do you want your identity to be public for this peer review?** For information about this choice, including consent withdrawal, please see our Privacy Policy .

Reviewer #1: No

Reviewer #2: No

---

## [Author Response · Author response to Decision Letter 1]

13 Nov 2024

Dear Academic Editor & Peer reviewers,

Thank you for your insightful and thorough comments, which we have used to strengthen and develop our manuscript: “Reading through the eyes of a university student: A double-masked randomised placebo-controlled cross-over protocol investigating coloured spectacle lens efficacy in adults with visual stress.”, for consideration of publication in PLOSOne. An edited document with highlighted changes to the text has been included in the re-submission. To improve readability, addressed comments from reviewer #1 are highlighted in purple text, with amendments made from reviewer #2’s comments are noted in red text. Where both reviewers raised the same point and an amendment was made, altered text has been written in green text. Detailed point-by-point response to each of the peer reviewers’ queries can be seen below, with reference to which line number(s) in the manuscript, which have been amended are also included.

Reviewer’s Comments to Authors

Reviewer 1:

The protocol is a longitudinal study of colored lenses in reading for university students. There are numerous statistical deficiencies due to the terseness of the protocol:

Author Response: Thank you for your comment in identifying potential statistical issues. We appreciate your comment and have amended the section on statistical analysis in lines 410 to 437.

1. The randomization procedure is not specified (permuted blocks, complete randomization, e.g.).

Author Response: Thank you for identifying this omission. We have now amended the manuscript on lines 275 to 278, to inform the reader that the randomisation procedure is a 1:1 randomisation without stratification. Further, we provide a rationale as to why this methodology was employed in the amended manuscript.

There is no discussion of the prevention of ascertainment or selection bias.

Author Response: Thank you for mentioning a potential source of bias that may develop during our trial. We have added a paragraph to our discussion, identifying the limitations of this trial. We have acknowledged the potential for ascertainment and selection bias and have discussed methods which we will employ to mitigate these issues. The discussion can be found on lines 485 to 501.

2. The sample size is based on a treatment difference seen in prior studies. Studies are powered for a CLINICALLY MEANINGFUL DIFFERENCE, not the observed effect size for prior studies.

3. No details are given on what sample size computations are used; since the outcome is based on a repeated measures ANOVA, there need to be measures of variability and covariance in order to compute the sample size. No indication is given as to the distribution of the primary outcome, which appears to be a count or a score based on words.

Author Response: Thank you for raising these points in relation to sample size calculations. We acknowledge that our initial sample size calculation may have been erroneous in nature and have re-calculated our data for a repeated measures ANOVA as our primary outcome measure. Using statistical software (G-Power, Version 3.19.4, 2024), we have calculated an a priori sample size for a within-factors repeated measures ANOVA. A partial eta squared (〖ηρ〗^2) of 0.0355 was used with a conservative small to medium effect size (f) of 0.1985. Probability of a type-1 error (α), of 0.05 and power (1-β) of 0.90 remained unchanged. As this is a crossover trial, only one analysable group are to be considered with 3-repeat measures of reading speed: at baseline, upon initial spectacle collection and after 6-weeks of spectacle wear. Correlation among repeat measures was assumed 0.5 and nonsphericity correction (ϵ) was held at 1.0. This calculation generated a sample size of n=59. These details have been included in lines 124 to 129 of the manuscript.

4. There are no details on the statistical modelling, what model will be used, if any covariates will be examined, what model assessment diagnostics will be used, what modelling assumptions will be made.

Author Response: Thank you for this consideration. We appreciate your suggestion to assess for covariates. Statistical differences in our primary outcome measures will be determined by repeated measures ANOVAs. However, we will consider the effects of covariates such as baseline reading speed, hours spent wearing spectacle and number of days wearing spectacles in post-hoc analysis through repeated measures ANCOVA testing, as required. This comment is addressed in lines 415 to 418. As this is a relatively homogenous population of undergraduate university students of similar age, other covariates are not considered relevant.

5. "Repeat" measures ANOVA should be "Repeated" measures ANOVA.

Author Response: Thank you for pointing out this mistake. We have amended the protocol and have changed the word ‘repeat’ to ‘repeated’ on line numbers 411, 414 and 429.

6. I don't understand the first part of the title, and the punctuation after "Reading".

Author Response: We thank the reviewer for this comment and apologise for making it difficult to understand. The first part of the title is used in all participant documentation within the study to allow a suitably vague title to prevent participants having prior knowledge of the protocol to avoid potential study bias. This title is also linked to the study protocol held on the Clinical Trials website (https://clinicaltrials.gov/study/NCT04318106?cond=visual%20stress&rank=7). We appreciate the first part of the title does not help inform the reader of the content of the publication but would be grateful it if could remain to link the Clinical Trials registration with the current protocol publication. We appreciate that the use of the semi-colon may be redundant. As such, we have removed this from the trial name. This change can be seen in line 4.

Reviewer 2:

Thank you for a well written protocol that helpfully covers most areas of the SPIRIT guidelines. I have several comments that you should consider to improve the manuscript.

Author Response: Thank you for your comprehensive and constructive comments. We have found this feedback highly beneficial at improving the strength of our manuscript. Individual changes to the manuscript are detailed below.

Introduction

The three paragraphs are do not follow a logical flow for me. I suggest: visual stress including prevalence and significance, aetiology and evidence supporting proposed mechanisms, treatment options including evidence for and against, professional guidelines, primary aim of trial.

Author Response: Thank you for your feedback. The introductory paragraphs have been re-drafted to follow the three areas suggested. The amended paragraphs occupy lines 64 to 102 of the manuscript.

You state “in small scale RCTs” and yet any RCT should be powered to find a difference i.e. scale does not matter. I suggest your statement is unpacked further. For example, do all trials show reduced symptoms and is this statistically and/or clinically significant?

Author Response: Thank you for highlighting this. We accept that stating “small scale RCTs” may not be appropriate and have re-worded this. We have now addressed both statistically and clinically significant differences in reading speeds in previous randomised control trials and how these vary greatly in a selection of previous studies reported in lines 92-96.

Study Design

What is your argument for a two-week washout period?

Author Response: We have now described the rationale for a 2-week washout period in the manuscript, included in lines 306 to 308. Two reasons are outlined. Firstly, this time period is equivalent to previous randomised control trials of precision tinted lenses, investigating precision tinted lens efficacy. Secondly, it is proposed that this is an appropriate interval to which colour memory for a chosen colour is negligible – as described further by D’Ath et al (2007).

References:

D’Ath P, Thomson W, Wilkins AJ. Memory for the color of non-monochromatic lights. Col. Res. Appl. 2007; 32: 11-15 DOI: https://doi.org/10.1002/col.20281

One concern here is that there must be some uncertainty in determining a tint that lies just outside the Limits of Therapeutic Effect but also appears similar enough for masking. It would be good to argue this clearly given some (limited) evidence from Suttle et al (2017) that results are variable.

Author Response: Thank you for this insightful comment. We acknowledge that there is a certain degree of variability in how we will ensure that the tint lies outside the Limits of Therapeutic Effect. We have expanded our methodology in lines 260 to 265, stating that the researchers will attempt to ensure that the control tint remains ‘inert’ through confirmation of participants’ visuoperceptual reading symptoms when viewing text under the identified control chromaticity. We appreciate that the work of Suttle et al (2017) suggests variability in results. One way in which the authors propose that variability in results may be minimised, is that all baseline data measures (including) Intuitive Colorimetry, will be conducted by a single examiner D.H., reducing any potential risk of interpractitioner variability, as discussed further in lines 324 to 327.

References:

Suttle CM, Barbur J, Conway ML. Coloured overlays and precision-tinted lenses: poor repeatability in a sample of adults and children diagnosed with visual stress. Ophthalmic Physiol. Opt. 2017; 37: 542-548 DOI: https://doi.org/10.1111/opo.12389

Study Population

Good to argue why you have chosen a student population because it would seem difficult to extrapolate to a wider population affecting the generalisability of your findings. For example, it is reasonable to anticipate an age range clustered around 18-24 with less participants as you go to your upper age limit of 45 (unless you are using stratified sampling). Also, the amount of reading your sample does will be higher than the general population.

Author Response: Thank you for this observation. We are in agreement that a student population may not be wholly representative of the wider adult population. Yet, researchers are confident that setting upper age-limit of 45-years, will avoid recruitment of participants experiencing reading discomfort as a result of age-related uncorrected refractive error – i.e. presbyopia. This is discussed further in lines 136 to 140. As this investigation is a superiority trial, our primary outcome measure is to determine if a clinically significant benefit can be endorsed with use of precision tinted lenses. Furthermore, this trial will explore coloured lens efficacy in a cohort of subjects with an equitable baseline level of reading competence, as all undergraduate students enrolled will have to exhibit demonstrable English language competence to enrol at Ulster University. If benefit of coloured lenses can be determined through this study, the authors appreciate that repeat investigations will be needed in generic/stratified populations, which may more accurately represent the adult population in the U.K or indeed a paediatric population. We reference these study limitations in lines 485 to 501.

Sample size Calculation

Did you assume a RM ANOVA when carrying out this power calculation? At a basic level the SD of the difference is fundamentally important rather than the within subjects SD, which could differ between groups. Could this be given?

Author Response: Thank you for this useful comment. As also addressed by reviewer 1, we have now adjusted our sample size calculation for repeated-measures ANOVA. We have re-calculated our data for a repeated measures ANOVA as our primary outcome measure. Using statistical software (G-Power, Version 3.19.4, 2024), we have calculated an a priori sample size for a within-factors repeated measures ANOVA. A partial eta squared (〖ηρ〗^2) of 0.0355 was used with a conservative small to medium effect size (f) of 0.1985. Probability of a type-1 error (α), of 0.05 and power (1-β) of 0.90 remained unchanged. As this is a crossover trial, only one analysable group are to be considered with 3-repeat measures of reading speed: at baseline; upon initial spectacle collection and after 6-weeks of spectacle wear. Correlation among repeat measures was assumed 0.5 and nonsphericity correction (ϵ) was held at 1.0. This calculation generated a sample size of n=59. These details have been included in lines 124 to 129 of the manuscript.

Recruitment

This section has sub-sections on screening tests and doesn’t comment on recruitment of participants!

Author Response: Thank you for this comment. We have now amended the manuscript in lines 219 to 222 to explain the recruitment process.

Outcome Measures

Please ensure the exact metric for the outcome measure is reported. And also indicate if it is an average or median measure for example and if any repeats are used.

Author Response: Thank you for your feedback. Average repeated measures ANOVA with and without coloured lenses will be conducted for the following reading tests, when assessing reading speed: Wilkins Rate of Reading Test; Adult Reading Test (2nd Edition); Test of Word Reading Efficiency; Developmental Eye Movement Test. As the pattern glare test can only be scored on an ordinal scale (i.e. possible pattern glare scores are limited to values of 0,1,2,3,4,5,6 and 7), median change in pattern glare test score will be reported. A test of normality (Shapiro-Wilk Test), will be conducted when conducting statistical analysis. In the event that data is non-normally distributed, non-parametric equivalents of the statistical tests employed will be assessed; namely a Friedman test in place of repeated measures ANOVA. As McNemar’s Test of Proportions is non-parametric in nature, this will not need to be adjusted. These amendments are included in lines 343 to 344, 410 to 415 and 433 to 434.

Why are you using a percentage reading speed change? Percentages are not normalised and relate to different effect size changes.

Author Response: We appreciate that percentages are not-normalised in nature. We will be reporting percentage change alongside whether there is a statistically significant change in Wilkins Rate of Reading Test Reading Speed by repeated measures ANOVA. Percentage change in reading speed has frequently been used to measure clinically-significant improvement in reading performance with/without coloured lenses and in recent investigation, has been proposed indicative and potentially diagnostic of visual stress – see Evans et al (2017). We accept the reviewer’s comment on the non-normalised nature of percentages and as such will amend our statistical plan to additionally report effect sizes. This amendment can be seen on lines 348 to 353.

References:

Evans BJW, Allen PM, Wilkins AJ. A Delphi study to develop practical diagnostic guidelines for visual stress (pattern-related visual stress). J. Optom. 2017;10: 161-168 DOI: 10.1016/j.optom.2016.08.002.

DEM

You state “will be noted and inform …”. By “inform” do you mean “calculate”? Best to be explicit.

Author Response: Thank you for this suggestion. The manuscript has been amended on line 369 to state that these metrics will be used to ‘calculate’ horizontal and vertical reading speeds.

Statistical Analysis

I was unclear about several points here that could usefully be clarified for the reader:

(1) Are you going to report effect sizes and confidence intervals? (good practice)

Author Response: Thank you for raising this point. On reflection and as discussed above, effect sizes and confidence intervals for change in reading speed on the Wilkins Rate of Reading Test with/without coloured lenses, will now be reported upon trial completion. This is discussed in lines 352 to 353.

(2) Are you going to adjust for baseline measures?

Author Response: Thank you for raising this query. We have updated the manuscript on lines 414 to 418 to state that we will consider baseline reading speed, the number of errors made in each reading test, number of days spectacle lenses worn, number of hours spectacle lenses worn and number of hours completing concentrated tasks with or without the use of coloured lenses, as covariates.

(3) Are you going to test for carry over and period effects?

This can

---

## [Decision Letter · Decision Letter 1]

27 Nov 2024

PONE-D-24-33918R1Reading through the eyes of a university student: A double-masked randomised placebo-controlled cross-over protocol investigating coloured spectacle lens efficacy in adults with visual stress.PLOS ONE

Dear Dr. Harkin,

Thank you for submitting your manuscript to PLOS ONE. After careful consideration, we feel that it has merit but does not fully meet PLOS ONE’s publication criteria as it currently stands. Therefore, we invite you to submit a revised version of the manuscript that addresses the points raised during the review process.

We look forward to receiving your revised manuscript.

Kind regards,

Clara Martínez Pérez

Academic Editor

PLOS ONE

Reviewers' comments:

Reviewer's Responses to Questions

**Comments to the Author**

1. Does the manuscript provide a valid rationale for the proposed study, with clearly identified and justified research questions?

Reviewer #1: Yes

2. Is the protocol technically sound and planned in a manner that will lead to a meaningful outcome and allow testing the stated hypotheses?

Reviewer #1: Yes

3. Is the methodology feasible and described in sufficient detail to allow the work to be replicable?

Reviewer #1: Yes

4. Have the authors described where all data underlying the findings will be made available when the study is complete?

Reviewer #1: Yes

5. Is the manuscript presented in an intelligible fashion and written in standard English?

Reviewer #1: Yes

6. Review Comments to the Author

You may also provide optional suggestions and comments to authors that they might find helpful in planning their study.

Reviewer #1: All comments have been addressed…………………xxxxxxxxxxxxxxxxxxxxxxxxxxxxxxxxxxxxxxxxxxxxxxxxxxxxxxxxxxxxx

7. PLOS authors have the option to publish the peer review history of their article (what does this mean? ). If published, this will include your full peer review and any attached files.

**Do you want your identity to be public for this peer review?** For information about this choice, including consent withdrawal, please see our Privacy Policy .

Reviewer #1: No

---

## [Author Response · Author response to Decision Letter 2]

9 Dec 2024

Dear reviewer and editor, following most recent confirmation from Academic Editor and Peer Reviewer(s), please find attached the most up-to-date copy of this manuscript. As there were no changes requested to this revision and all comments were addressed from the first revision, the manuscript remains unchanged from our most recent submission. Therefore, the rebuttal letter, marked-up copy and blank manuscript are still valid. and are attached in this submission. If you require any further assistance, please do not hesitate to contact me.

Many thanks for your assistance with this submission.

---

## [Decision Letter · Decision Letter 2]

17 Jan 2025

PONE-D-24-33918R2Reading through the eyes of a university student: A double-masked randomised placebo-controlled cross-over protocol investigating coloured spectacle lens efficacy in adults with visual stress.PLOS ONE

Dear Dr. Harkin,

Thank you for submitting your manuscript to PLOS ONE. After careful consideration, we feel that it has merit but does not fully meet PLOS ONE’s publication criteria as it currently stands. Therefore, we invite you to submit a revised version of the manuscript that addresses the points raised during the review process.

We look forward to receiving your revised manuscript.

Kind regards,

Clara Martínez Pérez

Academic Editor

PLOS ONE

Reviewers' comments:

Reviewer's Responses to Questions

**Comments to the Author**

1. Does the manuscript provide a valid rationale for the proposed study, with clearly identified and justified research questions?

Reviewer #1: Yes

Reviewer #3: Yes

2. Is the protocol technically sound and planned in a manner that will lead to a meaningful outcome and allow testing the stated hypotheses?

Reviewer #1: Yes

Reviewer #3: Partly

3. Is the methodology feasible and described in sufficient detail to allow the work to be replicable?

Reviewer #1: Yes

Reviewer #3: Yes

4. Have the authors described where all data underlying the findings will be made available when the study is complete?

Reviewer #1: Yes

Reviewer #3: No

5. Is the manuscript presented in an intelligible fashion and written in standard English?

Reviewer #1: Yes

Reviewer #3: Yes

6. Review Comments to the Author

You may also provide optional suggestions and comments to authors that they might find helpful in planning their study.

Reviewer #1: xxxxxxxxxxxxxxxxxxxxxxxxxxxxxxxxxxxxxxxxxxxxxxxxxxxxxxxxxxxxxxxxxxxxxxxxxxxxxxxxxxxxxxxxxxxxxxxxxxxxxxxxxxxxxxxxxxxxxxxxxxxxx

Reviewer #3: A good quality study on this research question is very much needed. As the authors point out, many practitioners test and prescribe coloured lenses for visual stress, despite very weak evidence for this practice. I applaud the authors for planning this study. However I have a number of concerns as outlined below.

Page 4 line 84: theories (i) and (ii) seem to be mixed up here; reduction in hyperexcitation relates to theory (ii).

It is worth noting that while the hyperexcitation theory has been used as the basis for experiments on coloured filters in migraine, autism, ADHD and other conditions as well as visual stress, the theory is not well founded. FMRI experiments (Huang et al, 2011, referenced in this manuscript) have shown that in people with visual stress, coloured filters reduce activation slightly in one or two higher visual areas but not in V1, which is surprising and has not been well explained. In addition, another study (Kim et al 2015) has found increased activation in visual stress, so this picture is not at all clear and this should be acknowledged when discussing possible hyperexcitation.

Page 5, line 96: there have been more than two systematic reviews that have concluded more RCTs are needed to establish whether coloured filters are beneficial in visual stress. The authors should review this again.

It is unclear how the control colour will be set – this should be specified more carefully.

Page 7 – near point of convergence ‘less than 10cm’ should probably be greater than 10cm – please check. The acuity criteria should apply monocularly and this should be specified. Reason for exclusion of photosensitive epilepsy should be explained. I suggest that as well as taking into account deviation such as tropia and decompensating phoria, fixation disparity at distance and near should also be considered, and vergence reserves should be measured (it has been indicated that stereopsis, associated phoria and vergence reserves are key measures – see Evans et al 1995, 1996). I suggest fixation disparity may also be a useful addition criterion since it may explain some symptoms.

Page 9 – it seems surprising that recruitment began two years ago. This being the case, has the study been underway for a while, and have the methods described here been used? If so, it may not be possible to apply the suggestions in this review. The purpose of a protocol is to declare methods before beginning data collection. Therefore if data collection has begun the purpose of this protocol seems unclear. Recruitment is to be completed by 20th December 24 (the date of writing this review) – this would be a long way into the students’ studies – if data collection is yet to begin, do they still want to participate, and if so the data collection would need to take place very soon if it is were to be completed while they are still at the University.

Further to the above, on page 23 it is stated that 60 participants have been recruited and are enrolled in the trial. Again, this seems surprising prior to the publication of a protocol. The point of an RCT protocol is to set out the methodology before beginning the study. That way the methods are followed and not modified as the trial progresses; if the researchers decide to change their methods they record this change in the protocol (e.g. Tetzlaff et al 2012 Guidelines for randomized clinical trial protocol content: A systematic review). Publishing the protocol after the study has begun seems to miss the point that a clear protocol guides conduct of the trial. The authors should consider and explain the point of publishing a protocol at this stage.

Page 10-11: The reading symptom score developed by the researchers seems to be key to visual stress diagnosis. As the authors point out, no clear system of diagnosis is available, but use of an unvalidated questionnaire will be a limitation of the study. If data collection has not yet begun, I suggest a pilot study should be conducted to test validity of this questionnaire.

Questions on symptoms while reading include whether the letters appear jumbled and whether lines are skipped and the participant is invited to use the foreign language text to help them answer. It seems that most text in a foreign language would appear jumbled and difficult to follow smoothly. Have the authors considered using a language the participant understands? A reference is given to support the use of foreign language text but the rationale is not clear. In addition, will the same text be used each time symptoms are assessed? If so, there may be a slight learning effect as the participant becomes more familiar with it.

Page 11: The determination of an arc of therapeutic effect is a new approach, to my knowledge, in which a range of similarly beneficial colours is identified. Again, this may be open to criticism unless validated by at least testing repeatability, and by establishing whether the control tint just outside of this arc is repeatedly not beneficial.

Page 17: I suggest it may be helpful to audio record the WRRT test to allow researchers to check the result after the event. It is very difficult to accurately record mistakes, progress and completed time simultaneously, and live assessment of reading speed without a checking mechanism is likely to be flawed in my view.

Page 24: a reference is needed to support the statement that over 300 practices have an intuitive colorimeter, and other statements of fact in this section.

7. PLOS authors have the option to publish the peer review history of their article (what does this mean? ). If published, this will include your full peer review and any attached files.

**Do you want your identity to be public for this peer review?** For information about this choice, including consent withdrawal, please see our Privacy Policy .

Reviewer #1: No

Reviewer #3: **Yes: ** Catherine Suttle

---

## [Author Response · Author response to Decision Letter 3]

11 Feb 2025

Reviewer #1:

We thank reviewer one for their time in reviewing the initial and revised version of this manuscript. We appreciate that no further comments have been raised by the reviewer. We wish to highlight their valuable contributions and we believe their insights have strengthened our manuscript.

Reviewer #3:

Reviewer #3: A good quality study on this research question is very much needed. As the authors point out, many practitioners test and prescribe coloured lenses for visual stress, despite very weak evidence for this practice. I applaud the authors for planning this study. However I have a number of concerns as outlined below.

Author Response: We thank reviewer #3 for their comprehensive and valuable comments, which we have used to strengthen our manuscript. We have addressed each of your concerns below individually. The manuscript has been marked up in red text where an amendment has been made in response to reviewer #3’s comments. Individual line number changes have been detailed in the comments below.

Page 4 line 84: theories (i) and (ii) seem to be mixed up here; reduction in hyperexcitation relates to theory (ii).

It is worth noting that while the hyperexcitation theory has been used as the basis for experiments on coloured filters in migraine, autism, ADHD and other conditions as well as visual stress, the theory is not well founded. FMRI experiments (Huang et al, 2011, referenced in this manuscript) have shown that in people with visual stress, coloured filters reduce activation slightly in one or two higher visual areas but not in V1, which is surprising and has not been well explained. In addition, another study (Kim et al 2015) has found increased activation in visual stress, so this picture is not at all clear and this should be acknowledged when discussing possible hyperexcitation.

Author response: We appreciate this comprehensive suggestion exploring the aetiological theories of visual stress. We apologise for mislabelling theories (i) and (ii). These have now been amended in lines 81 to 86. We accept the reviewer’s comments as to how we can provide a more nuanced response in relation to the theory of cortical hyperexcitability. We have expanded this theory in lines 87 to 94.

Page 5, line 96: there have been more than two systematic reviews that have concluded more RCTs are needed to establish whether coloured filters are beneficial in visual stress. The authors should review this again.

Author response: Thank you for bringing this to our attention. We have now included further reviews and meta-analyses in the introduction in lines 99 to 102 of the manuscript.

It is unclear how the control colour will be set – this should be specified more carefully.

Author response: We thank the reviewer for their comment. Control tint will be set outside the ‘Limit of Therapeutic Effect’ – an area of proposed beneficial chromaticities mapped on the 1976CIELUV chromaticity diagram. We propose that this therapeutic limit will be individualised for each observer. As detailed in previous investigations, there are wide variations in the size of Just Noticeable Differences (JNDs) between observers when assessed on the intuitive colorimeter.

The control tint being placed outside the Limit of Therapeutic Effect is set to such a point that we propose will be inert for the participant. At the upper and lower boundary of the Limit of Therapeutic Effect, participants will consider if reading symptoms / visual discomfort experienced are ‘worse’ or ‘no different’ compared no colour presented. When participants confirm that the presented hue is ‘no different’ compared to no colour (i.e. not beneficial nor aversive), researchers propose that this be a candidate for the control tint. As the therapeutic area will have upper and lower boundaries, there will be two candidate control tints. The control tint chosen will be set outside the therapeutic limit closest to the optimal tint, to assist with participant masking – i.e. the apparent colour difference will be more alike between control and optimal tint. The manuscript has been adapted on lines 281 to 287, detailing these amendments.

Page 7 – near point of convergence ‘less than 10cm’ should probably be greater than 10cm – please check.

Author response: Thank you for bringing this point to our attention. We were meaning that the poorer response would be an NPC distance greater than 10cm and we have amended this to ‘greater’ than 10cm as you suggest. This change is seen on line 166.

The acuity criteria should apply monocularly and this should be specified.

Author response: Exclusion criteria for monocular acuity (distance and near) have already been supplied in the manuscript – see lines 162 and 163.

Reason for exclusion of photosensitive epilepsy should be explained.

Author response: We have amended our exclusion criteria to explain the rationale behind diagnosis of photosensitive epilepsy as an exclusion criterion. This amendment is detailed in lines 171 to 173.

I suggest that as well as taking into account deviation such as tropia and decompensating phoria, fixation disparity at distance and near should also be considered, and vergence reserves should be measured (it has been indicated that stereopsis, associated phoria and vergence reserves are key measures – see Evans et al 1995, 1996). I suggest fixation disparity may also be a useful addition criterion since it may explain some symptoms.

Author response: We appreciate the comments made by the reviewer as to how we can enrich the binocular vision assessment of our participants prior to enrolment. Assessment of near associated phoria (by near Mallett unit) and measurement of positive, negative and vertical fusional reserves (measured by prism bar), will be conducted on all participants at baseline. Further binocular vision tests included at baseline include: calculation of the AC/A ratio (heterophoria method); accommodative facility (with +/-2.00D binocular flippers) and dynamic retinoscopy (with the Ulster-Cardiff cube). As the evidence base assessing the correlation between fusional reserves measures, associated phoria and visual stress is not definitive, we will not be using these tests to help determine exclusion criteria, but as further clinical tests conducted as part of a binocular vision work-up of each participant at baseline. An overview of the binocular vision work-up to be employed at baseline, has been detailed in lines 182 to 184 of the manuscript.

Page 9 – it seems surprising that recruitment began two years ago. This being the case, has the study been underway for a while, and have the methods described here been used? If so, it may not be possible to apply the suggestions in this review. The purpose of a protocol is to declare methods before beginning data collection. Therefore if data collection has begun the purpose of this protocol seems unclear. Recruitment is to be completed by 20th December 24 (the date of writing this review) – this would be a long way into the students’ studies – if data collection is yet to begin, do they still want to participate, and if so the data collection would need to take place very soon if it is were to be completed while they are still at the University.

Author Response: Owing to the relatively low assumed prevalence of visual stress in the adult population of ~15% and the large sample size required for this clinical trial, recurrent rounds of recruitment will be required in order to achieve sufficient participant recruitment numbers, if initial recruitment is sub-optimal. This amendment is seen on lines 223 to 225.

This investigation is seeking to ensure that our studied population is homogenous in nature, in order to draw meaningful conclusions from the outcomes of this trial. We wish to ensure our population is within a specific age range who all deal with a comparable level of reading demand – i.e. all participants are adult undergraduate students attending the same university in their first year of studies at time of initial trial recruitment.

We explore the reviewer’s reservations about the feasibility of publication in our next response. Yet, we can confirm that data collection has begun and participants have been motivated to take part. As the minimum length of time of an undergraduate course at Ulster University Coleraine is three years, all participants are still in attendance at university and therefore, trial completion is feasible within the remits of the timeframe of this study.

Further to the above, on page 23 it is stated that 60 participants have been recruited and are enrolled in the trial. Again, this seems surprising prior to the publication of a protocol. The point of an RCT protocol is to set out the methodology before beginning the study. That way the methods are followed and not modified as the trial progresses; if the researchers decide to change their methods they record this change in the protocol (e.g. Tetzlaff et al 2012 Guidelines for randomized clinical trial protocol content: A systematic review). Publishing the protocol after the study has begun seems to miss the point that a clear protocol guides conduct of the trial. The authors should consider and explain the point of publishing a protocol at this stage.

Author Response: When determining whether this protocol may be published in PLOSOne, we consulted the guidelines for publishing detailed by the journal: https://journals.plos.org/plosone/s/submission-guidelines#loc-study-protocols.

PLOSOne states that a clinical trial protocol must: (i) relate to a research study that has not yet generated results and (ii) be submitted before recruitment of participants or collection of data for the study is complete. We explore the reasons why our clinical protocol satisfy both criteria below:

(i) As this clinical trial is double-masked in nature and no statistical analysis shall take place until all participants have completed the trial, no results have been generated.

(ii) The clinical protocol has been submitted before recruitment of participants and collection of study data has been completed.

We would like to highlight that, to our knowledge, this will be the first clinical trial on precision tinted lenses to publish a clinical protocol in an academic journal, in advance of trial completion. We believe that this is of great importance for future academics to assist with replication of findings, if noteworthy in this investigation. Additionally, for context, we drafted and submitted this manuscript for publication some months ago, and have already been through a revision cycle.

Page 10-11: The reading symptom score developed by the researchers seems to be key to visual stress diagnosis. As the authors point out, no clear system of diagnosis is available, but use of an unvalidated questionnaire will be a limitation of the study. If data collection has not yet begun, I suggest a pilot study should be conducted to test validity of this questionnaire.

Author response: We agree with the reviewer, that currently no such validated visual stress questionnaire exists, and that the condition is diagnosed on an ad-hoc basis by eyecare practitioners. However, the items included in this reading symptom questionnaire were consolidated from a number of reading symptoms reported in the literature, proposed indicative of visual stress. We carried out further analyses on this questionnaire to assess its repeatability whilst screening large populations of students:

We conducted analyses into the test-retest reliability and internal consistency of the reading symptom questionnaire in order to confirm its validity. Cronbach’s alpha demonstrated high questionnaire internal consistency (α = 0.92). The questionnaire was retested on a subset of undergraduate students (n=115) after a period of 6-weeks. Retested participants were blinded to motivation for retesting as to avoid bias in responses. Furthermore, no participant with applicable reading symptoms was invited to partake in the clinical trial until after repeat testing had concluded, to reduce risk of biasing responses. Total summed symptom score two-way mixed average-measures intraclass correlation coefficient was deemed high (0.91). Bland-Altman analysis showed total summed questionnaire score bias was -1.99 (whereby a negative value indicated an increase on total summed score on repeat investigation). This suggests overall acceptable questionnaire test-retest reliability and internal consistency.

These findings have been briefly summarised in lines 199 to 206 of the manuscript.

References:

Bland JM, Altman DG. Statistical methods for assessing agreement between two methods of clinical measurement. Lancet. 1986;1: 307-310

Questions on symptoms while reading include whether the letters appear jumbled and whether lines are skipped and the participant is invited to use the foreign language text to help them answer. It seems that most text in a foreign language would appear jumbled and difficult to follow smoothly. Have the authors considered using a language the participant understands? A reference is given to support the use of foreign language text but the rationale is not clear. In addition, will the same text be used each time symptoms are assessed? If so, there may be a slight learning effect as the participant becomes more familiar with it.

Author response: To add some explanation of how the reading symptom questionnaire was structured, participants were asked to consider how their reading symptoms are when they read. As is detailed the supporting information in S2, the reading symptom questionnaire instructions do not explicitly ask participants to view the passage of foreign language text throughout the completion of the entire Likert scale section of the questionnaire (question one). The passage of crowded text is present to direct participants to when/if they struggle to answer a specific reading symptom question. The passage of crowded text acts as a visual aid for an uncomfortable ‘busy’ passage of text designed to evoke reading discomfort in symptomatic readers. This discomfort is probed specifically in question two of the questionnaire, where participants will be directed to view the passage of foreign text and document whether they do or do not experience any other reading symptoms when viewing the passage. Question two’s response was not considered as an inclusion criterion to the clinical trial, but will allow the researchers to assess the profile of symptomatic readers in post-hoc analysis as required.

To address the concerns related to potential learning effects of the foreign language passage; it is proposed that even with multiple exposures of written information through a unimodal input (i.e. solely visual input), foreign language vocabulary acquisition requires multiple exposures to new words in a meaningful context to acquire vocabulary (Bisson et al, 2014). We believe that the presentation of this unknown language to participants is not in a meaningful context and will reduce how likely participants are to acquire vocabulary, and therefore a learning effect be achieved. Participants were not directed to read the paragraph aloud nor was discussion surrounding the context of the foreign language passage facilitated, both of which are proposed to encourage vocabulary acquisition (Ulanoff & Pucci, 1999). Therefore, we believe that that any potential learning effects may be dampened by the fact that it will not be compulsory for participants to view the foreign language text upon repeat completions of the questionnaire.

References:

Bisson MJ, van Heuven WJB, Conklin K, Tunney, RJ. The Role of Repeated Exposure to Multimodal Input in Incidental Acquisition of Foreign Language Vocabulary. Lang. Learning: J. Res. Lang. Studies. 2014; 64(4): 855-877 DOI: 10.1111/lang.12085

Ulanoff SH, Pucci SL. Learning words from books: The effects of read aloud on second language vocabulary acquisition. Bilingual Res. J. 1999;23(4):409-422. DOI:

Page 11: The determination of an arc of therapeutic effect is a new approach, to my knowledge, in which a range of similarly beneficial colours is identified. Again, this may be open to criticism unless validated by at least testing repeatability, and by establishing whether the control tint just ou

---

## [Decision Letter · Decision Letter 3]

5 June 2025

Reading through the eyes of a university student: A double-masked randomised placebo-controlled cross-over protocol investigating coloured spectacle lens efficacy in adults with visual stress.

PONE-D-24-33918R3

Dear Dr. Harkin,

We’re pleased to inform you that your manuscript has been judged scientifically suitable for publication and will be formally accepted for publication once it meets all outstanding technical requirements.

Kind regards,

Clara Martínez Pérez

Academic Editor

PLOS ONE

Reviewers' comments:

Reviewer's Responses to Questions

**Comments to the Author**

1. Does the manuscript provide a valid rationale for the proposed study, with clearly identified and justified research questions?

Reviewer #1: Yes

Reviewer #3: Yes

2. Is the protocol technically sound and planned in a manner that will lead to a meaningful outcome and allow testing the stated hypotheses?

Reviewer #1: Yes

Reviewer #3: Yes

3. Is the methodology feasible and described in sufficient detail to allow the work to be replicable?

Reviewer #1: Yes

Reviewer #3: Yes

4. Have the authors described where all data underlying the findings will be made available when the study is complete?

Reviewer #1: Yes

Reviewer #3: No

5. Is the manuscript presented in an intelligible fashion and written in standard English?

Reviewer #1: Yes

Reviewer #3: Yes

6. Review Comments to the Author

You may also provide optional suggestions and comments to authors that they might find helpful in planning their study.

Reviewer #1: xxxxxxxxxxxxxxxxxxxxxxxxxxxxxxxxxxxxxxxxxxxxxxxxxxxxxxxxxxxxxxxxxxxxxxxxxxxxxxxxxxxxxxxxxxxxxxxxxxxxxxxxxxxxxxxx

Reviewer #3: One of my main concerns at the last round of review was that this is a protocol being published after data have been collected. The authors point out in their response that it is a PLOS One requirement that a protocol is not eligible for publication if results have already been obtained, and argue that they have gathered data but not obtained results since their data are masked. However, whether this is true depends on what PLOS One mean by results being obtained, and their requirements also refer to a published protocol 'sharing a study's design and analysis plan before the research has been conducted'. As far as I can tell this research has essentially been conducted. On re-reviewing the present protocol and the clinicaltrials.gov entry, it seems that the present protocol reflects the original study plan, but it is up to the journal editors to decide whether this protocol, describing a study at an advanced stage of data collection, satisfies the journal requirements as outlined above. If the protocol does satisfy journal requirements, the authors should at least explain briefly in the manuscript why it was not published at an earlier stage, prior to data gathering.

Other than the above, the authors have responded satisfactorily to my suggestions.

7. PLOS authors have the option to publish the peer review history of their article (what does this mean? ). If published, this will include your full peer review and any attached files.

**Do you want your identity to be public for this peer review?** For information about this choice, including consent withdrawal, please see our Privacy Policy .

Reviewer #1: No

Reviewer #3: No

---

## [Editor Report · Acceptance letter]

PONE-D-24-33918R3

PLOS ONE

Dear Dr. Harkin,

I'm pleased to inform you that your manuscript has been deemed suitable for publication in PLOS ONE. Congratulations! Your manuscript is now being handed over to our production team.

Kind regards,

on behalf of

Dr. Clara Martínez Pérez

Academic Editor

PLOS ONE